# Future Design as a Metacognitive Intervention for Presentism

**Yoshinori Nakagawa [1],\* and Tatsuyoshi Saijo [2,3]**

[1]   School of Economics and Management, Kochi University of Technology, Kochi 782-8502, Japan
[2]   Research Institute for Future Design, Kochi University of Technology, Kochi 780-8515, Japan;
      saijo.tatsuyoshi@kochi-tech.ac.jp
[3]   Research Institute for Humanity and Nature, Kyoto 603-8047, Japan
\*   Correspondence: nakagawa.yoshinori@kochi-tech.ac.jp

**Abstract:** Many serious problems occur due to conflicts between the interests of the present generation and the welfare of future generations, and thus, the actions of the preset generation may be a consequence of presentism. Drawing on the theoretical framework of metacognition, the present study investigates how presentism can be overcome through future design interventions that incorporate an imaginary future generation setting. Four workshop participants were interviewed, and transcripts of the interviews were made. There were two major findings. First, we identified narratives in the responses of participants that suggest that metacognition was active during the workshops concerning the two cognitions governed by present and future selves. Second, the narratives identified above were classified into two categories, and the two corresponding roles of metacognition were identified: the monitoring and controlling function and the harmonizing function. The former is essential for the acquisition of identity as a future person; the latter is essential for reconciling this future identity with the identity of the person in the present. The present study proposes that future design is a tool that can be used to intervene in the metacognition of individuals concerning how one chooses a temporal reference point from which to view the past, present, and future of society rather than a tool to naively motivate individuals to care for future generations.

**Keywords:** sustainability; future design; imaginary future generation; metacognition; qualitative research; sense of coherence

## 1. Introduction

Many serious problems currently threaten global sustainability, including climate change, environmental degradation, and national debt. These problems arise from conflicts between the interests of the present generation and the welfare of future generations, and thus, the actions of the present generation may be explained as a consequence of presentism, which is defined as the tendency to generate a bias within established laws and policies that have the potential to negatively affect future generations [1,2].

In order to overcome presentism within contemporary democratic systems, various attempts have been made to consider how the voices of future generations can be incorporated into current institutional decision making. A number of authors discussed the possibility of representing the interests of future generations through legislative measures [3–5]. Adachi [6] classified the institutionalization of the concerns of future generations into eight categories, including constitutional provisions for protecting the future generations' well-being [7], transferring the authority of some nation states to regional or global bodies [8], and creating an independent governmental agency to review all regulations expected to significantly impact future generations and oversee coordination among existing agencies [9].

In line with these studies, future design, a branch of future studies, has proposed a framework for conceptualizing the imaginary future generation [10,11]. Under this framework, individuals take on the role of members of an imaginary future generation and engage in designing strategies that can be adopted by the present generation. This design procedure can incorporate negotiation between the imaginary future generation and other participants who represent the perspective of the present generation in order to build intergenerational consensus. Concerned that the market and democratic institutions will continue to selfishly consume the resources of future generations, economist Tatsuyoshi Saijo proposed this framework as an alternative mechanism that can incorporate the interests and concerns of future generations in policymaking. Thus, while the framework of the imaginary future generation can be narrowly understood as a technique for operating workshops at organizations in order to create visions for the future, it should also be seen as a concept relevant to stakeholders who make various decisions in market-oriented democratic societies. A growing number of municipalities are utilizing this framework to hold workshops to generate visions for the future by inviting citizens to participate in workshops or organizing them in the municipalities' administrations (e.g., references [12,13]).

However, there is limited research on the subjective experience of individuals of the present generation taking on the role of members of a future generation. Previous studies confirm that the framework of the imaginary future generation encourages individuals to view sustainable policies more favorably despite inconveniences to the present generation [14–16]. However, it is important to answer the question of how identities of the present and future generations, which likely conflict with one another, can coexist in a single individual. The present qualitative study aims to answer this question.

As the issue of practical, real-life, institutional representation of future generations' voices remains unresolved, answering this question is more urgent than ever. In fact, drawing on case studies of institutions representing future generations in six countries and regions, Jones, O'Brien, and Ryan [17] (p. 158) concluded, "Institutions which are given too much power, too early in their lifespan, tend to face rejection from politicians." These cases clearly show how introducing representatives of the future generation into the contemporary political system induces intra-generational conflict and fails to attain its intended goals. Thus, institutional attempts to represent future generations must be accompanied by the attempts of each stakeholder (e.g., politicians and voters) to overcome their own biases related to presentism. Drawing on the theoretical framework of metacognition, the present study suggests a path forward in order to achieve this ultimate goal.

## 2. Theoretical Framework

Metacognition refers to higher-level cognition (i.e., a mental action or process of acquiring knowledge and understanding through thought, experience, and the senses) of targeting cognitions at the object level [18]. Metacognition has attracted a great deal of scholarly attention partly because it is essential for school children and adults to learn effectively. According to a review by Veenman et al. [19], the most common distinction in metacognition is between metacognitive knowledge and metacognitive skills. The former refers to a person's declarative knowledge about the interactions between a person, task, and strategy characteristics [20], while the latter refers to a person's procedural knowledge for regulating problem-solving and learning activities [21].

Procedural metacognition, the latter type of metacognition, includes the processes of metacognitive monitoring (i.e., subjective assessments of ongoing cognitive activities), and metacognitive control (i.e., the regulation of current cognitive activities) [18,22]. According to Roebers [23], monitoring includes questions such as "How much effort do I have to put into learning this material?"; "Did I sufficiently learn this material to remember the details later on?"; and "How sure am I that this answer is correct?" Controlling includes actions such as selecting material for review while studying, differentially allocating study time to learning material, and withdrawing answers or terminating a memory search.

The present study utilizes the concept of metacognition to understand how individuals encountering the perspectives of the future generation reconcile their present identities and the identities of future generations, which may conflict with one another. Therefore, our research objective is defined as follows:

**Objective 1:** To confirm if metacognition is activated in order to target cognitions operated by present selves and selves of future generations.

**Objective 2:** To investigate the function (or role) of this metacognition.

Considering the above-mentioned previous studies on metacognition that monitor and control other (i.e., lower-level) cognitions, it is hypothesized that a person activates his or her metacognition to monitor and control another form of cognition by considering questions such as, "Am I adopting (or should I adopt) the perspective of the present or the future generation when thinking about the state of society of 2050?" When one adopts the perspective of the present generation, one is looking 30 years into the future and imagining what technologies will be available and what future people will be thinking about, etc. On the other hand, when one adopts the perspective of the future generation, one is absorbed in the world he or she is imagining, similar to how someone becomes absorbed in a story world while reading a novel. Moreover, when adopting the perspective of a future generation, it is possible to either lament or appreciate specific actions taken by the present generation, such as responses to COVID-19 or $CO_2$ emission reduction efforts. (Due to the outbreak of COVID-19, the present generation may be even more motivated to adopt this perspective, as the COVID-19 pandemic has caused massive changes to the social structure. As such, people are now more sensitive to the fact that actions taken by the present generation will have serious consequence for the future generation.) This is similar to how the present generation assesses the actions of past generations (see earlier studies on retrospective assessment [24]). Thus, for Objective 1, we expect a positive answer, and for Objective 2, we expect that monitoring and controlling will be the main functions of metacognition. This will be qualitatively verified.

## 3. Materials and Methods

The present study is based on interviews conducted with four citizens who participated in future design workshops organized by two municipalities in Japan: Yahaba Town (Iwate Prefecture, Japan) and Uji City (Kyoto Prefecture, Japan).

Yahaba Town in Iwate Prefecture has a population of about 28,000. The town is located about a 20–30 min drive away from Morioka City, which has a population nearly 10 times larger than Yahaba Town. Therefore, many Yahaba residents work in Morioka City. Yahaba Town Hall held a series of unique workshops in order to craft a long-term vision for the town. The workshops were designed to have local citizens act as future residents of the town and consider what steps the town should take in order to realize a vision for the future. Two citizens who participated in these workshops, Mikiko and Miki (anonyms), agreed to participate in the interview survey. They are both females and were in their 40s at the time of the interviews. The interviews were conducted six months after the series of workshops was completed. For more details on the procedure and the outcomes of the workshops, see Hara et al. [12].

Uji City is located in the southern part of Kyoto Prefecture and has a population of 180,000. It is famous for its cultural properties, such as Byodoin Temple, which is a World Heritage Site, and Uji Tea. Many tourists visit this city on weekends. The city had been developed as a residential area since 1960, and the population has increased rapidly for some time; however, since 2000, the population has been declining. In addition, the percentage of the population aged 65 and over is expected to increase at a pace exceeding the national average. In response, Uji City Hall started developing a workshop as an unusual strategy for making local residents think about the future of the community. Two residents who participated in these workshops, Yumi and Satoshi (anonyms), agreed to be interviewed. Yumi is

a female who was in her 30s at the time of the interview, and Satoshi is a male who was in his 60s at the time of the interview. The interviews were conducted 12 months after the series of workshops finished.

In both workshops, participants were divided into groups consisting of approximately four members each. Both groups of workshop participants were asked to imagine that they had time-traveled to the future and were living there at their current ages. Participants in the Yahaba Town workshop were asked to imagine would their town would be like in 40 years; residents of Uji City were asked to imagine what their city would be like in 30 years. Then, in Yahaba Town, the participants were requested to give a general description of the town in this imagined future. In Uji City, the participants were requested to describe the state of the local communities of the future city they were asked to imagine. In both municipalities, a staff member facilitated each group discussion, and the objectives agreed were used to create a narrative of the future. In each group, participants sometimes made contradictory assumptions about the external environment surrounding the municipalities (e.g., socio-demographics). While participants tended to try to achieve a consensus, there was no guarantee that they perfectly shared assumptions. Even so, they were able to share the hope that a specific future world could be realized. This expectation seemed consistent with the organizational studies literature stating that the vision of an organization should be something that is not only "viewed as desirable by employees" but also "is unlikely to be changed by market or technology changes" [25].

The request that participants imagine themselves as living in the future at their current ages is important for the following reason. Future Design is intended to resolve intergenerational problems, and thus is primarily concerned with how to resolve conflicts between present individuals and other individuals in the future, rather than resolving conflicts between the same individuals who have aged into the future. Time-traveling 40 years into the future on the presumption that one will also age 40 years allows one to keep his or her current identity, which is undesirable for the purpose of this study. However, imagining time-traveling while remaining the same is a good (even if not perfect) and easily implementable simulation that allows one to feel absorbed in the future that will be inhabited by future generations. In fact, Miki explicitly mentioned in the interview how this constraint helped her to succeed in taking the perspective of the future generation. (See the sixth paragraph of Appendix B as well as the results section where this statement is cited.)

In interviews, the participants were requested to talk about (i) why they decided to participate in the workshops, (ii) how they and their group members managed to adopt the perspective of the future generation, and (iii) how the workshop experience influenced their lives. The interviews lasted 1–2 h. The interviews were recorded and later transcribed. The transcription amounted to 129 pages in total. The transcription was analyzed with reference to the theoretical framework introduced in Section 2. Specifically, interviewees' statements were reviewed one by one and checked for interviewees' adoption of metacognition. Then, the statements identified were classified into groups, and the functions of metacognition were considered for each group. The narratives of the four participants are summarized in story form in Appendices A–D, respectively.

## 4. Results

In the transcriptions of the four interviews, two functions of metacognition were identified, and they will be individually described below.

### 4.1. Monitoring and Controlling Function

As expected, the monitoring and controlling function of metacognition was identified. During the workshops, Miki carefully considered whether she was successful in adopting the perspective of the future generation, rather than maintaining her perspective as a member of the present generation. In the excerpt below, she describes her experience of imagining what it would be like to time travel to the future with her age unchanged, as requested by the organizer of the workshop (See the sixth paragraph of Appendix B).

**Miki:** Well, I started imagining the future as if I lived there as an 80-year-old female and thought about what future would interest me. I gradually began imagining that I was living there at the age I am now. This was probably because I felt it was a pity that such an interesting future would be realized only after I got so old.

This statement suggests that her metacognition is monitoring the process of cognition on the level of the object (i.e., the cognition as the present self or the cognition at the future self) and controlling herself so that the cognition of the future self becomes dominant. Additionally, in addition to her, the other group members also performed monitoring and controlling. In fact, soon after the start of the discussion, one of the elderly members began to express his opinion, stating "I will be dead by then . . . " However, once he began immersing himself in this future world and envisioning it more vividly, he did not repeat this remark. The next excerpt from Yumi shows that she was also monitoring her cognition at the object level.

**Yumi:** During the workshops, I was always questioning myself as to whether I was successful in playing the role as purely a member of the future generation or not. I was concerned that I was simply expressing my selfish ideas as a member of the present generation, pretending that I was doing so from the perspective of the future generation.

This monitoring process allowed her to arrive at the understanding of what it was like to play the role of a member of the future generation, as shown in the eighth paragraph of Appendix C. This monitoring and controlling function of metacognition was also observable in Satoshi's case, as shown in the following excerpt.

**Interviewer:** Please tell me the story of what happened after the series of workshops was over.

**Satoshi:** Yes. The series of four workshops ended. What I thought was that at the end of the series of workshops I at last came to understand what it was like to be a member of the future generation, and the ending of the workshops meant that I would sooner or later lose this sense of understanding and return to what I used to be before I got familiar with the idea.

In saying so, Satoshi compares the cognitions of his present and future selves and prioritizes the latter (in his monitoring process). Indeed, he decided to participate in a citizen network as a founding member in order to provide himself with opportunities to activate the cognition of the future self (in his controlling process). The network consists of nearly 20 citizens who participated in the workshops and has the aim of considering the future of the city from the perspective of the future generation.

*4.2. Harmonizing Function*

Another function of metacognition that was identified is the harmonization of object level cognitions. If a member of the present generation successfully experiences adopting the perspective of the future generation, but then reassumes his or her identity as a member of the present generation, he or she momentarily experiences the coexistence of two potentially conflicting identities. Such conflict cannot be avoided, especially when one is inclined to support actions that will come to fruition in the immediate future. However, Mikiko's narrative illustrates how these identities can be harmonized. We demonstrate that metacognition has the function of harmonizing these two identities (and thus the cognitions associated with the perspectives of the present and future generations). See the last paragraph of Appendix A for the context of the excerpt.

**Interviewer:** Now that the series of workshops is over, you have gone back to living your everyday life. Do you think about things as a member of the present generation, or do you recall the experiences of workshops when thinking about actions taken by the present generation?

**Mikiko:** Well, that is a difficult question to answer. After all, we succeeded in absorbing ourselves in a world 40 years in the future, and that is why I sometimes say to myself, "this should be done right now, otherwise it will be too late." This feeling springs up from me as a member of the present generation.

This statement suggests that there was an interplay between the cognition governed by the self as a member of the present generation, and the cognition governed by the self as a member of the

future generation. It also suggests that Mikiko was aware that the two cognitions were consistent with one another (i.e., the state described as the future generation and the actions to be implemented by the present generation are connected by a means-end relationship), and thus, there is metacognition enabling the perception that the two cognitions are in harmony with one another. This function of metacognition is also confirmed in the following excerpt from Satoshi. This statement also illustrates how the two cognitions are perceived as existing in harmony.

**Satoshi:** Some may argue that what the future should be like should be the choice of the future generation. However, there may not be time for that. Without taking action now, it may be impossible to realize a desirable future.

It should be noted that linking the two cognitions by the means–end relationship is not the only way to harmonize these cognitions. Yumi discovered a consistency between the two cognitions in a unique way: not by identifying the means-end relationship, but by identifying an essential similarity between the two cognitions (see the eighth paragraph of Appendix C). Yumi is well aware that the cognitions of present and future selves may well reach different conclusions with regard to what should be (have been) done in the present. According to her belief, this is because we, as the present generation, are bound by conventional thinking, which prevents us from realizing what we really want to do. She also believes that taking the perspective of the future generation frees us from such conventional thinking, and thus, there is no contradiction between cognitions by present and future selves.

## 5. Discussion

Considering the interview responses discussed above, our answers to the two research questions referred to in the research objectives section are summarized as follows:

Concerning the first objective, we identified narratives in participants' interview responses that suggest that metacognition was active in the two cognitions at the object level governed by the present and future selves. Concerning the second objective, the narratives identified above were classified into two categories, and two corresponding roles of metacognition were identified: the monitoring and controlling function and the harmonizing function. The former is essential for the acquisition of an identity as a future person, and the latter is essential for reconciling the identity of a future person with the identity of a present person. Figures 1 and 2 illustrate these two functions. Both of these figures illustrate a pair of two selves (i.e., present and future selves), as well as the third self-overviewing the two. The difference between these figures lies in the role of the third self. Specifically, an individual wishing to take the perspective of the future generation was concerned about whether they were successful in doing so, and thus the third self in Figure 1 monitored and controlled the transition process between the two selves. In contrast, once the individual experienced a successful transition, a new concern arose for the third self about whether the two selves were in harmony rather than contradicting one another.

There are several important things to note about these findings. First, the narratives demonstrate that the monitoring and controlling function of metacognition derived mainly from workshop participants' experiences during the Future Design workshops. In these workshops, participants were requested to activate an unusual cognition process by adopting the perspective of the future generation. Therefore, participants must monitor and control themselves in order to successfully adopt this perspective. Previous scholarship has recognized both the difficulty and importance of disengaging from the present in future studies workshops [26]. The present study adds to academic scholarship on this issue by demonstrating that transition from a present to future perspective is perceived by participants to be a discrete rather than continuous process and can only be achieved through careful monitoring and controlling. As this is not an easy task, it is inevitable that some participants are better at performing this task than others. However, very few earlier studies have paid attention to how this differs among individuals. With the findings of the present study in mind, future studies should investigate whether the ability to disengage from the present is associated with general metacognitive competence and skills (e.g., references [27–30]). Indeed, it is of practical importance for workshop

organizers to understand participants' individual characteristics and, when possible, place participants who have relatively low competence of disengaging from the present in groups with people of higher competence, so that the latter can assist the former by leading discussions.

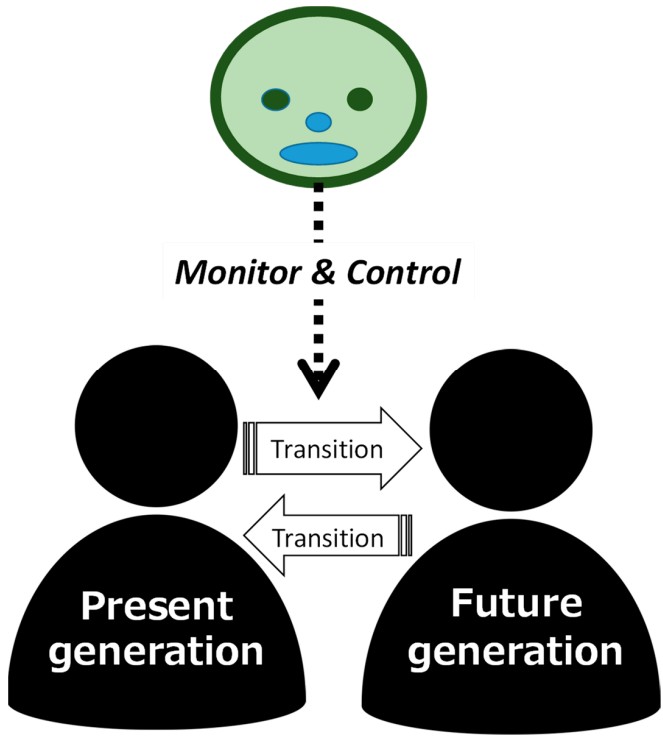

**Figure 1.** Illustration of the monitoring and controlling function of metacognition.

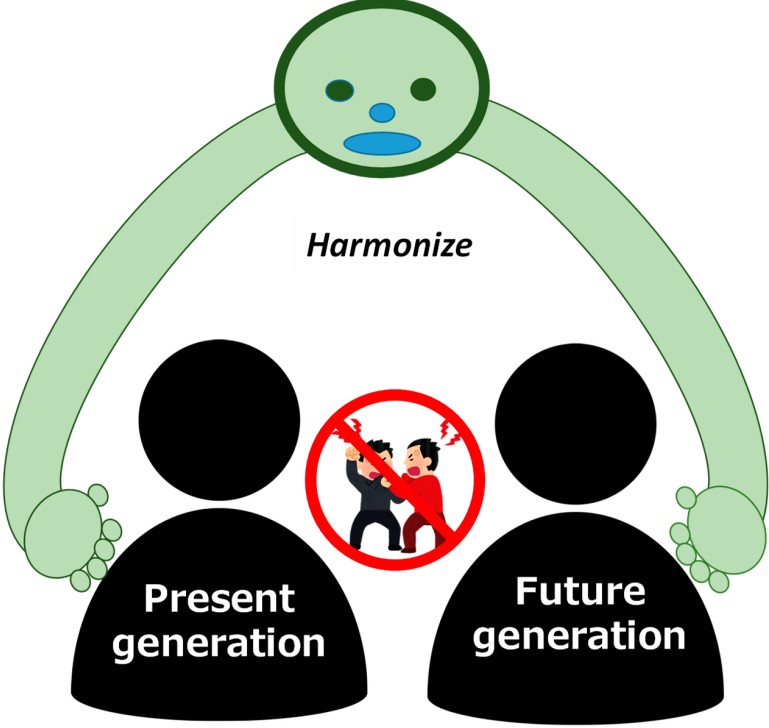

**Figure 2.** Illustration of the harmonizing function of metacognition (a part of this figure was borrowed from https://www.irasutoya.com.).

Second, in spite of potential conflicts of interest between present and future generations, as well as the difficulty of motivating the present generation to care for future generations [31], the present study found that the perspectives of the present and future generations can coexist on account of the harmonizing function of metacognition. Furthermore, it was found that this function can be executed in two different ways: either by defining the means-end relationship (i.e., between actions to be done by the present self and the state of the society in which the future self will live) or by identifying an essential similarity in spite of the apparent contradiction. This metacognition serves as a bridge between present and future selves. Therefore, metacognition is expected to contribute to one's sense of coherence [32,33] and thus one's life satisfaction.

Third, while the monitoring and controlling function of metacognition was active mainly during exposure to intervention (i.e., during the workshops where individuals are requested to take the perspective of the future generation), the harmonizing function of metacognition seemed to be active for a much longer period of time. In fact, Mikiko and Satoshi were interviewed six and 12 months, respectively, after the series of workshops finished, and even at that time, they still felt a sense of coherence, as demonstrated in the excerpts. In the future, researchers must quantitatively investigate the durability of these Future Design intervention. Answering this question will help determine whether Future Design is a viable and promising framework for realizing a sustainable society.

To conclude, regardless of the intentions of the organizers of the Future Design workshops in Yahaba Town and Uji City, the workshops can be viewed as interventions that triggered metacognition processes in the four participants, forcing them to consciously consider from what reference point in time they should evaluate the past, present, and future of society. Considered alongside the empirical evidence demonstrating the effectiveness of adopting the future reference point for engendering positive attitudes toward sustainable thinking [15,16], it would be beneficial for future research to consider how the metacognition of our four research participants can be replicated among citizens of contemporary society. Indeed, this would be preferable to naively questioning how many members of society as possible can be motivated to care about future generations. The former question seems to be less difficult to answer than the latter, because adoption of this form of metacognition is accompanied by a sense of coherence, which is an essential component of life satisfaction.

The present study had an important limitation. This qualitative study was based on interview surveys with four people only; therefore, in the future, it will be important to check the generalizability of the findings either by enlarging the sample size or by designing and implementing a questionnaire survey.

**Author Contributions:** Conceptualization, Y.N. and T.S.; Methodology, Y.N.; Analysis, Y.N.; Writing—Original Draft Preparation, Y.N.; Writing—Review & Editing, Y.N. and T.S. All authors have read and agreed to the published version of the manuscript.

**Funding:** This research received no external funding.

**Acknowledgments:** The authors are grateful to Keishiro Hara (Osaka University), who supported the series of Future Design workshops in Yahaba Town. We also wish to thank the four anonymous interviewees. The interview surveys were arranged by Ritsuji Yoshioka (Yahaba Town Hall personnel), Takayuki Sugimoto (Uji City Hall personnel), and Takayuki Onishi (Uji City Hall personnel).

**Conflicts of Interest:** The authors declare no conflict of interest.

## Appendix A. The Story of Mikiko in Yahaba Town

Mikiko is a cheerful and outspoken woman in her 40s. She was born and grew up in Iwate Prefecture. After graduating from a local high school, she went on to a university in a big city outside of the prefecture. After graduating from university, she considered staying in the big city to work. However, she decided to return to the prefecture she grew up in and work in Morioka City. After a while, she decided to get married. At first, she and her husband thought about living in Morioka City where they were working, but instead, they decided to build a house in Yahaba Town, a town that they

have no connection with. Despite the town being located about 20–30 min's drive away from Morioka City, they made this decision because the cost of building in the town is reasonable,

Ten years have passed since Mikiko move to Yahaba Town. She has few complaints about living in the town. Even though there is not much play equipment in a park near her house, it is sufficient for children to play. The elementary school that her children attend is also close to her home. Above all, the town's tap water tastes better than the tap water in Morioka City. The only unfortunate thing is that many young people who were born and raised in the town go on to universities outside of the prefecture and get jobs there. There are some households near her house whose sons and daughters moved outside the prefecture. Mikiko ended up participating in one of the unusual workshops.

On the first day of the workshop, each local participant, who had been assigned to groups of six, was invited into a room. There were various people in the room where Mikiko was invited, including a young woman around the age of 20 and an elderly man. The workshops were planned by Mr. Yoshioka, an employee of Yahaba Town Hall. At the beginning of the workshop, Mikiko could not understand the instructions. The instructions were as follows: "Imagine if you could time travel 40 years into the future to Yahaba Town and live there at your present age. Then, as a group, envision the town's future in a way that represents the interests of the future generation 40 years from now. Provide ideas for policy measures that we should implement now." Contrary to the instructions, Mikiko tried to envision the future 40 years from now, imaging that she would be an 80-year-old elderly woman, as she thought it was difficult to time travel 40 years into the future at her present age. Even so, she had thought that, "If I were an elderly woman in the future, I would be fine if the town would remained unchanged, wouldn't I?" As she has no particular complaints about living in Yahaba, it is natural that she had this thought.

First, the group held a discussion: which vision should they choose for their town 40 years from now? Should it be just as it is now, or different? During the discussion, one of the members of the group, an approximately 20-year-old young woman, expressed, "40 years later from now, I would like to hope that our town would be easier to live in than it is now." Her opinion itself was not that surprising. However, the young woman's remarks, which clearly expressed her hope that the group would choose a future vision of the town that was different from how it is not, affected Mikiko deeply. Indeed, it mad her think, "Well, yes, she is right." The young woman was born and grew up in Yahaba Town, and now commutes to a university in Morioka City while living in the town.

Triggered by the young woman's remark, Mikiko started to think that she also wanted to choose a different vision from the present. In 40 years, when the population of the town is aging, she wants it to be a place where young people who can support the elderly would think "We don't want to move outside of the town." The town surely should be like that. In fact, the town should be inundated with applicants hoping to move there from all parts of the country to the point where it would be inevitable to reject the majority of them. In this way, Mikiko and the group members expanded their imaginations. They decided that a playground for children and an exercise and health management facility for senior citizens should be established in the same place, so children and the elderly could come together. The facility should also serve as a workplace for the elderly to cook lunch boxes and delicatessens, and the young generation could buy those meals and communicate with the elderly. Thus, the group reached the concept of "Yahaba, a Healthy Town." This concept was found to be consistent with the plan of a medical university to establish a hospital in this town a few years after the workshops (which was established), as well as with other ideas of the members of the group, such as transforming the grounds of a closed junior high school into a sport stadium.

On another day of the workshop, a debate between the participants who adopted the perspective of the future generation (Mikiko's group who envisioned Yahaba's future as the future generation) and the participants of the present generations (the group who ordinarily lived in the present town as the present generations) was held. They discussed the policy measures that the town should implement going forward. The present generations insisted that important things to achieve were free medical care and education for children. When Mikiko heard their opinions, she found that members of the

present generation were obsessed with solving immediate issues. She appealed to the representatives of the present generation, stating, "We are living in a world 40 years in the future which is determined by the actions that you take now. Therefore, we will be at a loss if you fail to take proper action."

Even so, Mikiko didn't have the slightest intention of criticizing the present generations. If she was in the position of the present generations, she also would have focused on immediate issues facing residents now, and would have focused on solving these issues in a timely manner as needed, and would have seen the world 40 years in the future as a result of the accumulation of such short-sighted behavior. Therefore, in the debate, she would have insisted that we should solve immediate issues facing society now. Moreover, considering the future 40 years from now, she had even thought, "If I were an elderly woman in the future, I would be fine if the town remained unchanged as it is now, wouldn't I?" Indeed, her thinking only changed after she heard the idea of the female university student at the first workshop. For that reason, she understood that the present generation commonly criticizes the future generations, saying, "Consider more immediate issues more realistically!"

A few months after Mikiko had participated in the workshops, she reflected on her experience participating in the workshop. She feels that the experience taught her how to form her own opinions concerning what the citizens of the town need, and when. In fact, when she walks around the town and observes the scenery, sometimes, she remembers the days of the workshop and how she experienced the perspective of the future generation. This leads her to think, "We will be too late to act unless we do something immediately." Mikiko's perspective as a future person certainly still lives on in her mind. She is very grateful to Yahaba Town for giving her this valuable experience.

**Appendix B. The Story of Miki in Yahaba Town**

Miki lives in Yahaba Town with her husband and two children. One of her children is in their first year of junior high school, and the other is in their fourth year of elementary school. Six years ago, upon the invitation of her friend, she became a member of a civil panel that studies and evaluates the town's waterworks. At that time, she had been overwhelmed with childcare duties and household chores, even more than now, and had been just getting by day by day. However, her new responsibility allowed her to temporarily forget about childcare and allowed her to try something new and fresh. Without participating in the civil panel, she would never have had a chance to understand how water management works, such as the production of tap water and home water supply, which allow us to get clean water just by turning on a faucet. Miki ended up participating in this unusual workshop.

On the first day of the workshop, each local participant, who was each assigned to a group of six, was invited into a room. There were men and women of all ages in the room where Miki was invited. The workshops were planned by Mr. Yoshioka, an employee of Yahaba Town Hall. At the beginning of the workshop, Miki and the other participants received the following instructions: "Imagine if you could time travel 40 years into the future to Yahaba Town and live there at your current age. And then, as a group, envision the town's future in a manner that represents the interests of the future generations 40 years from now. Provide ideas for policy measures which we should implement now." However, since Miki lives in the present and had trouble imaging herself as a person in the future, she hesitated to participate in the group discussion. Contrary to the planners' instructions, she tried to imagine herself as an 80-year-old elderly woman living 40 years in the future, yet she was still apprehensive to share her thoughts with the group.

After the discussion had been going on for a while, the group began discussing how much of a priority the government was placing on childcare support, such as free medical care and daycare centers. This subject was relevant to Miki, as she was overwhelmed with childcare duties and felt that it was a crucial issue. Contrary to her thoughts, Mr. Omura, a 70-year-old member of the group, expressed his opinion. "Since we are living in the future, we can assume that such issues will have been solved in a few years from now, and we won't have to deal with that problem 40 years from now."

Miki was taken aback by his casual comment, stating, "I had been thinking of issues that I wanted to be solved in the near future, and therefore I had been limited by thinking about childcare too much.

That is why I could not imagine being a future person. Issues like childcare will likely be solved in a few years, and it will surely be solved 40 years from now. I'll temporarily forget about daily childcare. I'll temporarily put aside issues which will likely be solved in a few years, and I'll imagine Yahaba town 40 years from now!"

After that, Miki began thinking about what kind of town would be interesting to live in when she is 80 years old, and she started leading the workshop discussion. One of the ideas reached by the group was to create a public transportation network to connect schools, the Town Hall, and tourist spots, adopting the slogan "Night on the Galactic Railroad," a children's story by Kenji Miyazawa. Future vehicles could possibly connect those places. Giovanni, the main character of this story, gets on the Galactic Train from a small hill with his one friend and finds what true happiness is while traveling through various constellations. Giovanni wakes up when he loses sight of his friend and finds out that all of the events that happened were in his dream world. In the real world, he notices that his friend fell into the river to save his acquaintance and went missing. Mt. Nansho in Yahaba is said to be the model for the hill where the train starts in this beautiful story.

In this way, while Miki was envisioning Yahaba's future 40 years from now and imagining herself at 80 years old, she felt as if she actually lived in the future world at her present age. She noticed that other members of the group also shared the same sensation. In fact, soon after the start of the discussion, one of the elderly members began one of his comments by saying, "I would be dead by then . . . " However, once he began immersing himself in imaging this future world, he no longer made such a remark. In this way, the members of the group gradually and naturally came to follow the instructions by Mr. Yoshioka, who said, "Imagine if you could time travel into the future at your present ages."

On a later day of the workshop, a debate between the participants representing the future generation (Miki's group) and participants representing the present generation (the group who ordinarily lived in the present town as the present generation) was held. They discussed the policy measures that the town should implement going forward. As expected, the present generation insisted that the town should implement immediate measures such as childcare support. At that time, Miki felt, "They are still talking about such issues (laughing). Such issues already have been solved in our future world (laughing)." However, this does not mean that she was looking down on the representatives of the present generation. Before becoming part of the group representing the future generation, she also insisted that immediate issues be dealt with first. If she and the members representing the future generation had instead been assigned to the group representing the present generation, they likely would have thought that the vision of a future transportation network (the "Galactic Railroad") was an empty dream. Therefore, once Miki experience the perspective of the future generation, she likely felt that her thinking had matured.

Now that the series of workshops is over, Miki thinks back to the remark of Mr. Omura, a 70-year-old man, which allowed her to assume the perspective of a future person. She felt that she could respond to Mr. Omura because she both relies on and is attached to her country (Japan). Working mothers face many challenges, such as chronic shortages of daycare centers and the persistently low involvement of men in childcare in Japan. However, the Japanese must not be so foolish as to leave such issues unsolved for 40 years. The Japanese surely will be able to solve such issues in the near future. That is why Miki realized that finding solutions to such problems is the responsibility of the present generation and was able to put such issues aside and imagine time-traveling 40 years into the future. Acting as a future person was a beneficial experience for her.

**Appendix C. The Story of Yumi in Uji City**

Yumi is a woman in her 30s and serves on a PTA (parent–teacher association) board at the elementary school attended by her first-born, 10-year-old daughter. When her daughter reached the age to enter elementary school, a problem arose. Her daughter started saying, "I never want to go to elementary school." Since then, she is going to school but regularly doesn't attend despite being

enrolled, and now she is in the fourth grade. Some odd things often occur. When her daughter is absent from school, she goes to her daughter's school for PTA activities.

When her daughter became unable to go to school, she noticed something. There were only mothers with their little children and elderly people strolling around the streets in the daytime during weekdays. Yumi and her elementary school daughter stood out in such a situation, so sometimes elderly strangers at a park said things to her like, "Children should go to school properly." This caused her to feel somewhat ashamed. She fully understood the meaning of their advice. However, she doesn't want to force her daughter to go to school and ignore her daughter's feelings. She wants her daughter to go to school according to her own will and say things like "I want to go to school" or "I want to study." She also wants her to absorb many things.

Yumi knows that most people cannot accept such opinions about school. When she has such thoughts, she often feels like her heart will break. When she felt this way, the ability to share her feelings with other mothers who have children in a similar situation was an import form of support for her. Each mom lives in a different area, but they send text messages by mobile phone to encourage each other. Yumi, in such a situation, happened to see a leaflet to recruit participants for the unusual workshop.

The idea for the workshop was interesting: "Try to become a future person and consider what the local community of Uji City will be like in 30 years." On the day of the workshop, there were many elderly people who resembled the elderly people who spoke to her at the park. This made her feel a little uneasy, but her anxiety gradually decreased during a discussion with a group of four people. The instructions given the participants by the City Hall personnel were as follows: "Imagine if you could time travel 30 years into the future to Uji City and live there at your current age. And then, tell us what the city's local community would look like."

At first, she felt ashamed to express her opinion while imagining to be a future person. When she forgetfully said something like, "The other day, I watched a program on TV," the facilitator would correct her, saying, "You mean that TV program was on 30 years ago, right?" However, thanks to the staff member's constant corrections, she was able to adopt the perspective of a future person without being held back by her thoughts from the present. In expressing her ideas about the future, she talked about children being able to attend schools in different school districts and take classes across grade boundaries. She realized that no one in the group rejected her opinion. She was also surprised and pleased with the group's warm-hearted attitude and the feeling of togetherness she felt between the elderly and herself.

She was also surprised that many of the elderly participants in the groups representing the future generation also imagined new visions of schools. One group envisioned a future in which children could study special fields without divisions between grade levels, such as elementary, junior high, and senior high schools. Another group envisioned a future in which children could study and take credits at home, and each student could develop their ability on their own volition. In Japan, at present, when children turn six years old, they must enter elementary school at the same time and then go to junior high and senior high in that order. This standardized school system continues to exist. However, many groups representing the future generation quickly and easily abandoned such conventional thinking.

She felt hopeful when she saw that many participants envisioned free styles of learning as future persons. As soon as the workshop was over, she sent messages to her mom friends with whom she always exchanged emails over mobile phones, saying, "Now the workshop is over. Contrary to our expectations, all the elderly might have the same hopes as we have!"

There are many issues to be solved that will affect the next generation. In the process of starting an action, we may meet people who hold different opinions from us. Yumi thinks it is too one-sided to say to such people, "You should change your opinion." However, we can at least share "things that we want to cherish," which is normally put in the back of our minds due to restrictions imposed by social relationships and conventional thinking in daily life. For her, the experience of being a future person

impelled her to temporarily abandon conventional thinking, which had been restricting her usual life, and freely envision a world that she wants to live in.

In the spring, a few months after she experienced being a future person, her daughter became a fourth grader in elementary school. Although Yumi has a child who doesn't like the school, she decided to become a PTA board member because she wanted to contribute to making the school better for all children (even if just a little) so that they can think of school as "incredibly enjoyable." She wants parents and guardians and local people to understand the benefits of thinking about things from the perspective of a future person. This hope was expressed clearly in a newsletter message which she wrote as a PTA board member.

### Appendix D. The Story of Satoshi in Uji City

There is a tea house on the approach to Byodoin Temple. The tea house is on the first floor of a Japanese-style modern building that was built using bare concrete and wooden materials. The building harmonizes with the surrounding buildings while also being unique. Satoshi, a main character of this story, is serving tea to customers while talking cheerfully at the tea counter. He is in a good mood when customers are attracted to the building and enter the tea house. The reason is because he designed the building and has an attachment to it.

Actually, Satoshi is not the owner of the tea house. He runs an architectural design office on the second floor of this building. When the tea house is crowded with customers, he comes down the stairs to help. He designed the building when he was 38 years old and ran a design office outside Uji City. He wanted to build a building with a distinctive presence and make the approach to the temple more attractive by having the building harmonize with the other shops. That was his wish.

When Satoshi was 50 years old, he had a chance to move his design office to the second floor of the building. After that, he became the chairman of an association of 60 members, including shop owners and neighbors located around the front approach to the temple. The members cannot always agree on every issue, including how to regulate traffic during tourist peak hours or how to receive newcomers.

Satoshi happened to get a leaflet recruiting participants for a series of citizen workshops. He was attracted by the theme of the workshops, Future Design, which has a positive meaning, as it is associated with designing and creating the future. After the participants were divided into groups of four at the workshop, they were given the following instructions by the City Hall employee: "Imagine that you could time travel 30 years into the future to Uji City and live there are your present age. Then, envision what the city's local community would look like."

For members of Satoshi's group, this task was more difficult than they expected. They had trouble imagining themselves in a future 30 years from now, and envisioning Uji in the future, as well. The reason was that they tended to discuss solutions for the problems of the present, such as how to best use parks in the city. However, one of the other groups was imagining a future where people fly with Take-copters (Hopters). When he saw this, he thought, "They have become true future people," and was envious of them.

This experience made him think that imagining the future as a future person means considering things that people of the present cannot even imagine. So, for example, he had the following thought. The youth in Uji City are reluctant to join neighborhood and community associations. Therefore, instead of coming up with a simple stopgap measure, it would be better to come up with an exciting plan that could naturally bring the youth and elderly together.

One of the other groups envisioned a future in which the local elderly would set up a small private school at home to provide a learning place for children. By establishing such places, a bond between people would be built naturally. Consequently, many issues that are caused by the poor membership in neighborhood and community associations would also be resolved automatically. This idea taught Satoshi that we can envision an exciting future if we simply adapt new ways of thinking. Indeed, we don't need to imagine a dreamlike future with technology such as Take-copters.

Now that the series of workshops is over, Satoshi is still doing his best to apply this approach as the chairman of the association. He realizes that participation of the workshops improved his thinking process. The present area in front of the temple has many issues to solve, including regulating vehicles and pedestrians, and how to receive newcomers. However, attempting to solve these issues with stopgap measures is not enough. It is necessary that everyone can agree on a future vision, saying "We want to approach this problem in this way." By doing so, the present issues can be solved naturally.

What then is "the approach like that we in the association want?" The exact answer is not yet known, but the approach would be something where "people can consider the whole of Uji City as attractive due to the presence of the approach." When Satoshi was 38 years old, he thought, "I wish to make the whole front approach to the temple more attractive by constructing one building." He is now 62 years old and is looking forward to the future of Uji City as a whole.

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
