# Peer review of "Future Design as a Metacognitive Intervention for Presentism"

_sustainability, doi:10.3390/su12187552_

Round 1

Reviewer 1 Report

The article present a topic with is really interesting and actual. Some comments following the article structure:

Introduction: the need to overcome presentism is acceptable however, given the current situation determined by the pandemic - cited along the article - requires that in the introduction the important role that this event is playing is perhaps emphasized. In fact, not only does it represent an unprecedented element of discontinuity after the Second World War, but it focuses precisely on crucial issues for the future on which the article wishes to reflect.

Methology and tools: here the author/s must make the effort to explain better both the organization and the structure of the workshop. Even if the aim of the article is to explain and understand the metacognitive effort of the participant, the reader need a metacognive framework to understand the situation. 1) The content of the workshop is not clear: yes they act as future persons but which are the topics discussed? Health? Social Cohesion? Family? Work? Welness? Communication? Homes and/or city stucture? Services? Transport? 2) How the two participants are guided to imagine futureS (always plural!) ? Are they aware of the plurality of futures? Who are the other participants? Who they represents? How are guided the interactions during the workshops? Are there any facilitators? which is the output that they have to produce? 

A better explanation of the structure of the activities is needed and then we can focalize on how the research question where addressed during the experiment and later interviews.

I suggest that the explanation of the "ages" influence on the experiment must be explained in one single and separate paragraph also supported by the reference to the interviews. So to clarify the importance of this detail and its consequences. 

Discussion and results: when talking about metacognition it is important to explain also trough images the different level involved. Actually the pictures presented here are of little help. It would be important to crystallize your results in something visible and remarkable. I would suggest the author to consider a couple of works by Rivka Oxman and Cunluffy that talks of both metacognition and design often visualizing pretty well the process and the different positions of the actors involved.[ see: Cunliffe L. (1999), Learning how to learn, Art education and the background, in Journal of Art and Design Education, Vol. 18, p.115-121; Oxman, R., (2004), Think-maps: teaching design thinking in design education, in Design Studies 25, p.63–91]

Author Response

Comment 1. (Reviewer #1)

The article present a topic with is really interesting and actual. Some comments following the article structure:
Thank you very much for your appreciation.

Comment 2. (Reviewer #1)

Introduction: the need to overcome presentism is acceptable however, given the current situation determined by the pandemic - cited along the article - requires that in the introduction the important role that this event is playing is perhaps emphasized. In fact, not only does it represent an unprecedented element of discontinuity after the Second World War, but it focuses precisely on crucial issues for the future on which the article wishes to reflect.
Thank you for the suggestion. We inserted the following sentence.

(Due to the outbreak of COVID-19, the present generation may be even more motivated to adopt this perspective, as the COVID-19 pandemic has caused massive changes to the social structure. As such, people are now more sensitive to the fact that actions taken by the present generation will have serious consequence for the future generation.)

Comment 3. (Reviewer #1)

Methology and tools: here the author/s must make the effort to explain better both the organization and the structure of the workshop. Even if the aim of the article is to explain and understand the metacognitive effort of the participant, the reader need a metacognive framework to understand the situation. 1) The content of the workshop is not clear: yes they act as future persons but which are the topics discussed? Health? Social Cohesion? Family? Work? Welness? Communication? Homes and/or city stucture? Services? Transport? 2) How the two participants are guided to imagine futureS (always plural!) ? Are they aware of the plurality of futures? Who are the other participants? Who they represents? How are guided the interactions during the workshops? Are there any facilitators? which is the output that they have to produce?
Thank you for the suggestions. We modified the manuscript in the following ways. First, in the case of Yahaba, the authors were not engaged with the workshops at all, and thus it seems inappropriate to provide details in this study. Therefore, we inserted the following sentence:

For more details on the procedure and the outcomes of the workshops, see Hara et al. [12].

Second, this said, it is necessary to make explicit the theme of the group discussions in both of the municipalities, the following sentences were inserted:

Then, in Yahaba Town, the participants were requested to give a general description of the town in this imagined future. In Uji City, the participants were requested to describe the state of the local communities of the future city they were asked to imagine. In both municipalities, a staff member facilitated each group discussion, and the objectives agreed were used to create a narrative of the future. Voices were recorded and analyzed after the discussion. In each group, participants sometimes made contradictory assumptions about the external environment surrounding the municipalities (e.g., socio-demographics). While participants tended to try to achieve a consensus, there was no guarantee that they perfectly shared assumptions. Even so, they were able to share the hope that a specific future world could be realized. This expectation seemed consistent with the organizational studies literature stating that the vision of an organization should be something that is not only “viewed as desirable by employees” but also “is unlikely to be changed by market or technology changes” [33].

33. Kantabutra, S.; Avery, G.C. The power of vision: statements that resonate. J. Bus Strategy. 2010, 31(1), 37—45.

Comment 4. (Reviewer #1)

A better explanation of the structure of the activities is needed and then we can focalize on how the research question where addressed during the experiment and later interviews.
Thank you for the suggestion. The following sentence was added.

Specifically, interviewees’ statements were reviewed one by one and checked for interviewees’ adoption of metacognition. Then, the statements identified were classified into groups, and the functions of metacognition were considered for each group.

Comment 5. (Reviewer #1)

I suggest that the explanation of the "ages" influence on the experiment must be explained in one single and separate paragraph also supported by the reference to the interviews. So to clarify the importance of this detail and its consequences.

Thank you for the suggestion. We separated the sentences as an independent paragraph. Also, the following sentences were inserted:

In fact, Miki explicitly mentioned in the interview how this constraint helped her to succeed in taking the perspective of the future generation. See the sixth paragraph of Appendix B as well as the results section where this statement is cited.

Comment 6. (Reviewer #1)

Discussion and results: when talking about metacognition it is important to explain also trough images the different level involved. Actually the pictures presented here are of little help. It would be important to crystallize your results in something visible and remarkable. I would suggest the author to consider a couple of works by Rivka Oxman and Cunluffy that talks of both metacognition and design often visualizing pretty well the process and the different positions of the actors involved.[ see: Cunliffe L. (1999), Learning how to learn, Art education and the background, in Journal of Art and Design Education, Vol. 18, p.115-121; Oxman, R., (2004), Think-maps: teaching design thinking in design education, in Design Studies 25, p.63–91]

Thank you for the suggestion. Our immature way of presenting the figures made it difficult to interpret them. Thus, rather than directly replying to this comment by citing the above-mentioned studies, please allow us to reply to this comment by inserting the following sentences to explain the figures.

Both of these figures illustrate a pair of two selves (i.e., present and future selves), as well as the third self-overviewing the two. The difference between these figures lies in the role of the third self. Specifically, an individual wishing to take the perspective of the future generation was concerned about whether they were successful in doing so, and thus the third self in Figure 1 monitored and controlled the transition process between the two selves. In contrast, once the individual experienced a successful transition, a new concern arose for the third self about whether the two selves were in harmony rather than contradicting one another.

Reviewer 2 Report

I think this is a wonderful new way to approach Future Design. The paper is clear, if a little lengthy (the appendix, notably) and needs important corrections of the English language. I have tried to make suggestions in this direction, which are appended. The methodology is as far as I can see quite novel and yields interesting questions about humans' ways of thinking about present and future. However, there are for the moment only a very few cases - the approach urgently needs application to larger groups of individuals.

Author Response

Comment 1. (Reviewer #2)

I think this is a wonderful new way to approach Future Design. The paper is clear, if a little lengthy (the appendix, notably) and needs important corrections of the English language. I have tried to make suggestions in this direction, which are appended. The methodology is as far as I can see quite novel and yields interesting questions about humans' ways of thinking about present and future. However, there are for the moment only a very few cases - the approach urgently needs application to larger groups of individuals.

Thank you for the appreciation. At the same time, we deeply apologize for having you read our grammatically immature manuscript. We understand how carefully you read the manuscript, in spite of our carelessness. We will never forget your kindness.

After reflecting all of your corrections, we had the revised manuscript edited by an English language editing company. Also we inserted the following paragraph:

The present study had an important limitation. This qualitative study was based on interview surveys with four people only; therefore, in the future, it will be important to check the generalizability of the findings either by enlarging the sample size or by designing and implementing a questionnaire survey.